# Content of Phenolic Compounds and Organic Acids in the Flowers of Selected *Tulipa gesneriana* Cultivars

**DOI:** 10.3390/molecules25235627

**Published:** 2020-11-29

**Authors:** Agnieszka Krzymińska, Monika Gąsecka, Zuzanna Magdziak

**Affiliations:** 1Department of Ornamental Plants, Dendrology and Pomology, Poznań University of Life Sciences, Dąbrowskiego 159, 60-594 Poznań, Poland; 2Department of Chemistry, Poznań University of Life Sciences, Wojska Polskiego 75, 60-625 Poznań, Poland; monika.gasecka@up.poznan.pl (M.G.); zuzanna.magdziak@up.poznan.pl (Z.M.)

**Keywords:** chemical composition, edible flowers, tulip

## Abstract

The study focused on the determination of phenolic acids, flavonoids and organic acids in five tulip cultivars ‘Barcelona’, ‘Columbus’, ‘Strong Gold’, ‘Super Parrot’ and ‘Tropicana’. The cultivars grown in field and in a greenhouse were exposed after cutting to different times of storage (0, 3 and 6 days). The phenolic profile contained 4-hydroxybenzoic, 2,5-dihydroxybenzoic, gallic, vanillic, syringic, salicylic, protocatechuic, *trans*-cinnamic, *p*-coumaric, caffeic, ferulic, chlorogenic and sinapic acids, as well as quercetin, rutin, luteonin, catechin and vitexin. The mean phenolic acid content was in the following order: ‘Columbus’ and ‘Tropicana’ > ’Barcelona’ > ’Strong Gold’ > ’Super Parrot’, while the levels of flavonoids were as follows: ‘Strong Gold’ > ’Barcelona’ > ’Tropicana’ > ’Columbus’ > ’Super Parrot’. The highest content of phenolic acids was confirmed for Columbus and Tropicana, while the lowest was for Super Parrot. However total phenolic content was very similar, observed between the place of cultivation, time of storage and cultivars. Malonic, succinic, acetic and citric acids were the major organic acid components in tulip petals. More organic acids (except malonic) were accumulated in tulip petals from fields than those from the greenhouse, while changes during storage were strictly correlated with cultivars.

## 1. Introduction

The food sector is currently largely focused on natural products with health-promoting properties and aesthetic potential, which is why interest in flowers in the food sector is increasing significantly. During opening flowers biochemical, physiological and structural changes take place [1]. Chemical changes that occur during flower development can affect their nutritional and beneficial health properties [2]. However, one more aspect should be noted, i.e., different growth conditions as well as the time from picking to consumption may also affect the chemical composition of edible flowers. Phenolic compounds are secondary metabolites recognised in edible flowers as bioactive compounds. Phenolic metabolites are important antioxidants responsible for anti-inflammatory, anticarcinogenic and antibacterial properties and determining the colour of flowers [3]. Depending on the number of phenolic subunits, flower phenolic compounds belong to simple and highly polymerised phenols. Among them, flavonoids, phenolic acids and tannins are the most important in the human diet [4].

Moreover, different plant species and cultivars develop their own bio-chemical and nutritional characteristics, ultimately resulting in their unique quality [5]. This includes the formation of organic acid metabolic traits, which are critical to taste, because the fruit/flower development and storage are always accompanied with organic acid degradation [6]. Playing a crucial role in human metabolism, organic acids are beneficial to human health and thus highly recommended for a healthy diet [7]. However, such information is scarce for edible flowers. There are also very few publications discussing how chemical characteristics of edible flowers depend on their place of growth and time of storage. Identifying changes in the chemical and nutritional composition of edible flowers during their cultivation and storage would be a step to improve their quality and bioactive potential, and thus their nutritional and commercial value. Tulips are popular ornamental plants in gardens, flowering in April or May (northern hemisphere). Plants forced in greenhouses produce cut flowers throughout the year.

In recent years, there has been renewed interest in using flowers of ornamental plants for culinary purposes. Flowers were already eaten in ancient Greece and Rome. But it is only recently that the chemical composition and biological value of flowers has become a research subject. Also tulip flowers are edible. The chemical composition of tulip flowers was first analysed as early as the mid-19th century [8]. They were found to contain e.g., anthocyanins, carotenoids and several flavonols [9]. Thanks to their biochemical content, tulip flowers exhibit biological activities, such as antipyretic, anticancer, laxative, expectorant and depurative [10]. The chemical composition depends on the cultivar and the flower colour [9].

Edible flowers are harvested during the peak of bloom [11]. Keeping flowers in cold storage between 4–6 °C before packing can increase their durability for 2–5 days [12]. The aim of this study was to determine the content of phenolic compounds (phenolic acids and flavonoids) and organic acids in flowers of five tulip cultivars (different in terms of flower colour) grown in the field and in the greenhouse and stored for different times.

## 2. Results and Discussion

Edible flowers contain many bioactive compounds including phenolic compounds [3]. Among phenolic compounds flavonoids and phenolic acids are distinguished as the main non-flavonoid compounds [4]. In this study only petals were analysed because of the bitter taste of the other flower parts. Total phenolic content (TPC) ranged from 11.11 to 12.47 mg GAE g^−1^ FW, with generally no significant differences found between the cultivars, types of cultivation and storage times (Table 1). However, the red and yellow ‘Tropicana’ showed significantly higher mean content that the pink-coloured ‘Barcelona’. TPC of ‘Barcelona’ from greenhouse on the day of harvest was significantly lower than TPC of this cultivar subjected to the other combinations. Significant TPC differences were observed within a given cultivar besides ‘Super Parrot’ (Figure 1). Just like with other edible flowers, the range of total phenolic content in the studied tulip cultivars was very wide [13].

Profiling the extract from the tulip petals showed that they contained both flavonoids and phenolic acids. In the flavonoid profile many groups were identified, such as flavones (luteonin), flavonols (quercetin, rutin), flavan-3-ols (catechin) and an apigenin flavone glucoside (vitexin) (Table 1). The order of cultivars in terms of the mean flavonoid content was as follows: ‘Strong Gold’ > ’Barcelona’ > ’Tropicana’ > ’Columbus’ > ’Super Parrot’. Rutin was the most abundant flavonoid in the profile of nearly all cultivars harvested from the greenhouse and additionally in ‘Super Parrot’ from field cultivation on the day of harvest. The same applies to the mean rutin content in each cultivar. Catechin was dominant in ‘Columbus’, ‘Strong Gold’ and ‘Tropicana’ from field cultivation on the day of harvest. Storage induced changes in the content of these compounds. Only catechin content increased during storage. However, no changes were observed for the mean sum of flavonoids between the day of harvest and the 3rd day of storage. Rutin, quercetin and luteolin were found at higher levels for greenhouse cultivation, while for catechin and vitexin it was for cultivation in the field. The mean sum of flavonoids was higher in flowers from field cultivation. The same was observed for total acid content. Flavonoids are well known for their beneficial effect on human health [14]. In edible plants, flavonoids, or rather their subclass, flavonols, are the main phenolic metabolites [3]. Quercetin and rutin are important flavonols exhibiting anti-inflammatory and antiobesity properties [15]. Other types of flavonoids, such as catechin, have an effect on gut microbiota [16]. Some studies have confirmed that the presence of flavonoids is reflected in the colour of flowers, for example, the colours yellow and orange are related to the presence of flavonoids in flowers of edible plants [17]. Phenolic compounds are highly efficient antioxidants protecting other biomolecules from effects of oxidative damage.

Among phenolic acids, derivatives of hydroxybenzoic and hydroxycinnamic acids were determined both with the C6-C1 structure (4-hydroxybenzoic (4-HBA), 2,5-hydroxybenzoic (2,5-DHBA), gallic, vanillic, syringic, salicylic and protocatechuic acids) and the C6-C3 structure (*trans*–cinnamic, *p*-coumaric, caffeic, ferulic, chlorogenic and sinapic acids) (Table 2). Their content depended on the cultivar, type of cultivation and time of storage. The range of phenolic acid content was very wide: from 0.10 (salicylic acid for ‘Barcelona’, field, 3rd day of storage) up to 26.17 µg g^−1^ FW (trans-cinnamic acid for ‘Barcelona’ greenhouse, 3rd day of storage). Trans–cinnamic acid was the dominant phenolic acid in ‘Barcelona’ (field and greenhouse cultivation), ‘Columbus’ (greenhouse), ‘Strong Gold’ (greenhouse), ‘Super Parrot’ (field and greenhouse) and ‘Tropicana’ (greenhouse). For ‘Columbus’ (field) and ‘Tropicana’ (field) synaptic acid was the main phenolic acid, while protocatechuic acid was dominant in ‘Strong Gold’ (field).

The analysis of mean phenolic content showed that ‘Barcelona’, ‘Columbus’, ‘Strong Gold’ and ‘Tropicana’ contained the highest amounts of trans–cinnamic acid, while protocatechuic acid was most abundant in ‘Tropicana’. The mean content of each phenolic acid depended on the cultivar, with the highest content of 4-HBA, 2,5-DHBA and sinapic acids identified in ‘Barcelona’, the highest content of syringic, trans–cinnamic and chlorogenic acids found in ‘Columbus’, gallic and caffeic acids in ‘Strong Gold’, while protocatechuic, vanillic, ferulic and salicylic acids was dominant in ‘Tropicana’. Phenolic acids were also determined in other edible flowers [3]. The cultivars richest in phenolic acids (sum of phenolic acids) on the day of harvest were ‘Columbus’ (field) and ‘Tropicana’ (field and greenhouse), followed by ‘Barcelona’ (greenhouse). Comparison of the content of selected compounds in tulip flower petals with selected edible plants can be concluded that petals can have high nutritional properties in human nutrition. They contain comparable amounts, e.g., *p*-coumaric and chlorogenic acid to apples and mushrooms [18,19].

The type of cultivation significantly determined the content of phenolic acids. Generally, the sum of those acids (for each experimental system and as sum of acids content mean total) was greater in flowers from the greenhouse that from the field. The exception were ‘Super Parrot’ and ‘Tropicana’, in which case the sum of phenolic acids was higher in flowers grown in the field. In flowers from greenhouse cultivation gallic, protocatechuic, *trans*–cinnamic, caffeic, chlorogenic, ferullic and salicylic acids were recorded at higher mean levels than other acids. The content of 4-HBA, vanillic, syringic, 2.5-DHBA, caffeic and sinapic acids was higher in flowers grown in the field than in the greenhouse.

On the 3rd and 6th day of storage changes in the content of phenolic acids were noted. The mean content of protocatechuic, 4-HBA, vanillin, trans–cinnamic, 2,5-DHBA, caffeic, chlorogenic, ferulic and sinapic acids was the highest on the day of harvest and decreased during storage. By contrast, the content of syringic, *p*-coumaric and salicylic acids rose during storage. On the following days of storage, the phenolic acid profile underwent quantitative changes. Generally, total acid content (sum) decreased after storage, except for ‘Barcelona’ flowers grown in the field. Variations in the content of some phenolic acids depending on the flower colour were reported by Zheng et al. [20]. In another study, caffeic, chlorogenic and vanillic acids were found to improve lipid metabolism thus preventing obesity [21]. *p*-Coumaric acid exhibits antimicrobial and anti-inflammatory activity, as shown by Pei et al. [22] and, similarly as ferulic and sinapic acids, improved gut microbiota [23]. Sinapic acid exhibits antioxidative and anti-inflammatory properties [24].

The studied samples of tulip petals presented very distinct organic acid profiles (Table 3). Succinic, acetic, malonic and quinic acids were found to be dominant in all tulip cultivars; however, the sum of studied organic acids was specific to cultivar and time of storage. The largest amounts of all studied organic acids were found in the petals of ‘Barcelona’ on the 3rd day of storage and in ‘Super Parrot’ on the 6th day of storage - both from field cultivation. This result is commonly associated with the presence of acetic, malonic and succinic acids in ‘Barcelona’, as well as succinic, citric and malonic acids in ‘Super Parrot’. Also the content of quinic acid was relatively high in both cultivars grown in the field for ‘Barcelona’ and ‘Super Parrot’. However, in the case of flowers from greenhouse cultivation the highest content of organic acids was observed in ‘Tropicana’, where the main acid was malonic acid. It should also be noted that, as in the case of phenolic compounds, the content of organic acids significantly varied depending on how the tulips were cultivated. The sum of organic acids in tulip flowers was greater in flowers grown in the field than in those grown in the greenhouse. Field crops were generally found to have higher concentrations of oxalic, citric and lactic acids than of other acids. Oxalic, malic and fumaric acids were only found in trace amounts in the analysed tulips petals.

Few studies indicate that organic acids may have a protective role against various diseases thanks to their antioxidant activity (e.g., tartaric, malic, citric or succinic acids), capable of chelating metals or delocalising the electron charge coming from free radicals [25]. Of the exchangeable health-promoting acids, [26] showed high organic acid content in dried dahlia petals, mainly due to the presence of succinic and malic acids, while Miguel et al. [27] indicated the actual content of citric acid in marigold, with a simultaneous lack of succinic acid. In turn, Dias et al. [28] also showed that malic acid was the dominant acid in dandelion and dahlia petals. Each of those flower species was characterised by a specific content of organic acids, which could have resulted from the cultivar diversity, different stages of flower development, drying of the plant material, or—as in our research—the place of cultivation and storage time. It has been shown that tulip petals were a good and balanced source of organic acids, which in terms of the fact that these compounds have not been studied so far in edible tulip petals makes this study novel on the chemical characteristics of this plant. It has also provided a preliminary analysis of changes in the content of bioactive compounds in the petals of selected tulip cultivars. Further investigation is needed to clarify potential applications of tulips and as a result, phytochemical characteristics of the most bioactive molecules, such as phenolic compounds and organic acids, could be determined and correlated to its biological properties in order to better understand the attributes of this plant species.

## 3. Materials and Methods

### 3.1. Characteristics of Experimental Materials

Five tulip cultivars were analysed: ‘Barcelona’, ‘Columbus’, ‘Strong Gold’, ‘Super Parrot’ and ‘Tropicana’. Among them, the best known is ‘Strong Gold’. It is grown in the Netherlands, on approx. 10% of the country’s area cropped to a total of 1800 tulip cultivars. It has yellow flowers. Another important cultivar is ‘Columbus’ with full pink-cream flowers. The other cultivars have single flowers, pink in ‘Barcelona’, red-yellow in ‘Tropicana’ and white in ‘Super Parrot’.

Flowers were obtained from the greenhouse and field cultivation of Bogdan Królik farm in Chrzypsko Wielkie, Western Poland. For greenhouse production bulbs were planted in boxes with peat substrate at the beginning of October 2018 and were placed in a dark, low-temperature storage room with an initial temperature of 9 °C, which was gradually reduced to 2 °C until the end of the cold period. After 16 weeks boxes with bulbs were placed in the greenhouse at a temperature of 15–16 °C. For field production bulbs were planted in the middle of October 2018 in open field. Flowers from the greenhouse were collected on 4 March 2019, while those from the field—on 28 April 2019. Flowers were divided into three batches. The first batch was analysed on the day of harvest, the second after 3 days, while the last after 6 days. Flowers were stored at 4 °C.

### 3.2. Experiment Design, Sample Preparation

Phenolic compounds and organic acids were extracted from homogenised flower petals of each cultivar using 80% ethanol, sonicated for 20 min and then shaking for 12 h at room temperature. Then the samples were centrifuged at 3000 rpm, evaporated to dryness and stored at −20 °C before analyses.

### 3.3. Determination of Phenolic Compounds and Organic Acids

Phenolic compounds and organic acids were identified using Ultra-Performance Liquid Chromatography (ACQUITY UPLC H-Class System, Waters Corporation, Milford, MA, USA). The separation was archived on an Acquity UPLC BEH C18 column (2.1 mm, 150 mm, 1.7 µm, Waters) thermostated at 35 °C. The flow rate of 0.4 mL^.^min^−1^ and the gradient elution with water and acetonitrile (both containing 0.1% formic acid, pH = 2) were used. The identification of peaks was based on a comparison with the retention times of chemical standards. The detection was performed in a Waters Photodiode Array Detector set at λ = 280 nm and λ = 320 nm using an external standard.

### 3.4. Determination of Total Phenolic Content (TPC)

The flower extracts was mixed with the Folin-Ciocalteu reagent diluted with deionised water (1:1) and then the mixture was left at room temperature for 3min, after which 20% Na_2_CO_3_ was added. The samples were incubated for 30min at room temperature and absorbance was measured at 765 nm using a UV-spectrophotometer. Gallic acid was used as the standard for TPC quantification. The concentration of TPC was expressed as milligram gallic acid equivalents per fresh weight (mg GAE g^−1^ FW).

### 3.5. Chemicals

The standards of phenolic compounds (gallic acid, protocatechuic acid, 4-hydroxybenzoic acid, 2,5–dihydroxybenzoic acid, vanillic acid, syringic acid, catechin, caffeic acid, *p*-coumaric acid, ferulic acid, chlorogenic acid, sinapic acid, rutin, *trans*-cinnamic acid, quercetin, tuteolin, naringenin, apigenin, kaempferol, vitexin) as well as organic acids (acetic, citric, fumaric, lactic, malic, maleic, malonic, oxalic, quinic and succinic) were purchased in Sigma—Aldrich (Saint Louis, MO, USA).

### 3.6. Statistical Analysis

The factors of the experiment were the cultivar, the type of flower cultivation (greenhouse, field) and the time of storage (0, 3 and 6 days). There were 3 replicates in each experiment combination. The obtained results were subjected to three-factor variance analysis. The means were grouped using the Duncan test at significance level α = 0.05. Additionally one-factor analysis was conducted for total phenolic content, separately for each cultivar.

## 4. Conclusions

The study demonstrated phenolic and organic acid composition of tulip flowers. Phenolic profile was consist of phenolic acids (derivatives of hydroxybenzoic and hydroxycinnamic acids) and flavonoids. Their content depended on the cultivar, type of cultivation and time of storage. However the total phenolic content was significantly different only between the ‘Barcelona’ and ‘Tropicana’ cultivars. The place of cultivation and the time of storage had no effect of total phenolic content. More organic acids (expect malonic) were accumulated in tulip petals from the field than in those from the greenhouse. Changes during storage of organic acids were strictly correlated with cultivars. Forcing tulips in greenhouse can be used to extend the availability of edible flowers beyond the flowering date in the field production.

## Figures and Tables

**Figure 1 molecules-25-05627-f001:**
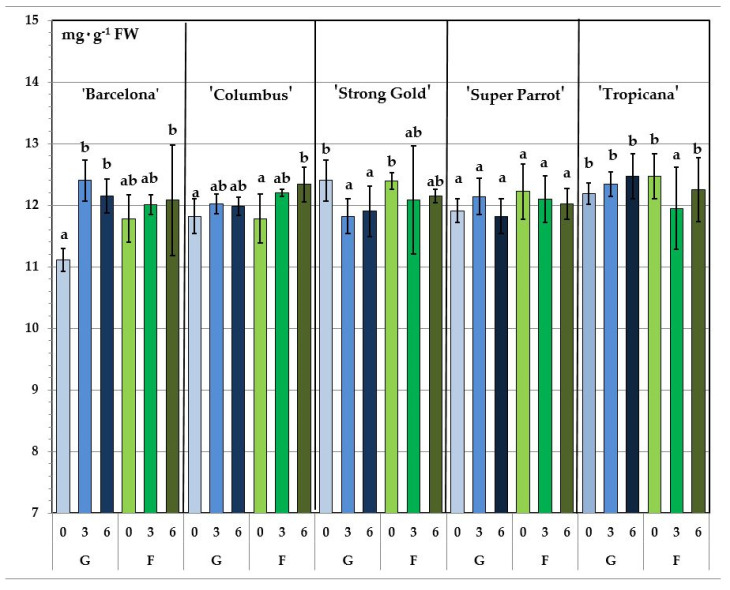
Total phenolic content of tulip flowers depends on cultivar. 0, 3, 6—time of storage (days). G, F—type of cultivation (greenhouse, field).

**Table 1 molecules-25-05627-t001:** Profiling of tulip flower flavonoids.

Cultivar (A)	Type of Cultivation (B)	Storage Length (Days) (C)	TPC		Katechin		Vitexin		Rutin		Quercetin		Luteolin		Sum	
			[mg GAE g^−1^ FW]		[µg g^−1^ FW]											
Barcelona	greenhouse	0	11.11	±0.19 a *	2.18	±0.09 f–i	1.69	±0.24 g,h	14.63	±2.42 i	0.05	±0.01 a,b	0.1	±0.01 c–j	18.66	±2.71 m
		3	12.4	±0.33 b	1.83	±0.22 d–h	2.25	±0.22 i	15.62	±2.26 i	0.35	±0.03 g,h	0.08	±0.01 a–g	20.13	±2.81 m
		6	12.15	±0.27 b	2.82	±0.33 j	1.21	±0.22 d–f	0.92	±0.21 a,b	0.03	±0.01 a,b	0.09	±0.01 a–h	5.07	±0.36 b,c
	field	0	11.78	±0.38 b	2.29	±0.22 h–j	5.04	±0.14 m	1.44	±0.20 a–d	0.3	±0.071 f–h	0.11	±0.01 i–k	9.18	±0.14 f–i
		3	12.01	±0.60 b	3.38	±0.44 k	4.43	±0.50 l	1.42	±0.68 a–d	0.09	±0.01 a–c	0.08	±0.01 a–c	9.39	±1.27 g–j
		6	12.08	±0.89 b	8.93	±0.59 o	1.73	±0.17 h	1.48	±0.14 a–d	0.07	±0.01 a–c	0.08	±0.01 a–g	12.29	±0.83 k
Columbus	greenhouse	0	11.82	±0.28 b	1.7	±0.18 c–h	0.29	±0.02 a	5.46	±1.29 f	0.42	±0.23 h,i	0.1	±0.01 e–k	7.96	±1.29 d–h
		3	12.02	±0.16 b	3.37	±0.14 k	1.58	±0.20 e–h	10.24	±1.35 h	0.13	±0.03 a–d	0.14	±0.01 l,m	15.46	±1.44 l
		6	11.98	±0.15 b	2.28	±0.38 h–j	0.73	±0.10 b,c	4.88	±0.57 e,f	0.2	±0.02 c–f	0.09	±0.01 a–h	8.17	±0.78 e–h
	field	0	11.78	±0.40 b	4.76	±0.51 m	1.28	±0.35 d–g	2.54	±0.24 b–d	0.24	±0.03 d–g	0.08	±0.01 a–d	8.89	±0.86 f–i
		3	12.2	±0.62 b	1.26	±0.36 b–d	1.75	±0.33 h	2.07	±0.49 a–d	0.12	±0.03 a–d	0.08	±0.01 a–e	5.27	±0.90 b,c
		6	12.34	±0.28 b	2.51	±0.22 i,j	0.46	±0.03 a,b	2.66	±0.19 b–d	0.09	±0.01 a–c	0.07	±0.01 a	5.79	±0.28 b–d
Strong Gold	greenhouse	0	12.4	±0.33 b	1.99	±0.11 e–i	0.91	±0.04 c,d	22.03	±2.67 j	0.13	±0.01 a–d	0.09	±0.01 b–i	25.14	±2.73 n
		3	11.82	±0.28 b	0.84	±0.15 b	1.29	±0.11 d–g	42.39	±0.59 k	0.14	±0.03 a–e	0.08	±0.01 a–f	44.75	±0.69 o
		6	11.9	±0.41 b	1.63	±0.27 c–f	1.26	±0.18 d–g	47.68	±2.09 l	0.13	±0.02 a–e	0.12	±0.05 j–l	50.82	±2.27 p
	field	0	12.39	±0.36 b	5.62	±0.47 n	1.34	±0.18 e–h	2.53	±0.41 b–d	0.25	±0.02 d–g	0.1	±0.01 f–k	9.84	±0.92 h–j
		3	12.09	±0.88 b	1.48	±0.40 c–e	0.7	±0.09 b,c	2.61	±0.32 b–d	0.02	±0.00 a	0.07	±0.02 a–c	4.89	±0.65 b,c
		6	12.15	±0.11 b	4.17	±0.27 l	1.19	±0.24 d,e	2.67	±0.04 b–d	0.12	±0.16 a–d	0.07	±0.01 a,b	8.23	±0.85 e–h
Super Parrot	greenhouse	0	11.91	±0.19 b	1.27	±0.28 b–d	1.4	±0.41 e–h	3.4	±0.42 d,e	0.37	±0.03 g,h	0.12	±0.01 kl	6.56	±0.36 b–e
		3	12.14	±0.30 b	0.71	±0.11 b	0.1	±0.03 a	1.01	±0.05 a–c	0.03	±0.01 a	0.1	±0.01 h–k	1.96	±0.07 a
		6	11.82	±0.28 b	0.19	±0.03 a	0.35	±0.05 a,b	1.29	±0.20 a–c	0.14	±0.01 a–e	0.11	±0.01 i–k	2.09	±0.14 a
	field	0	12.22	±0.44 b	2.25	±0.39 g–j	2.32	±0.28 i	2.58	±0.38 b–d	0.17	±0.03 b–f	0.08	±0.01 a–e	7.4	±0.30 d–g
		3	12.1	±0.37 b	3.5	±0.28 k	1.17	±0.16 d,e	3	±0.12 c,d	0.08	±0.01 a–c	0.09	±0.01 c–i	7.84	±0.28 d–h
		6	12.02	±0.25 b	1.16	±0.10 b,c	1.63	±0.10 f–h	1.78	±0.15 a–d	0.13	±0.01 a–e	0.08	±0.01 a–e	4.79	±0.10 b
Tropicana	greenhouse	0	12.19	±0.18 b	1.67	±0.27 c–g	0.28	±0.06 a	8.66	±1.62 g,h	0.26	±0.23 e–g	0.1	±0.01 d–k	10.96	±1.80 i–k
		3	12.34	±0.20 b	1.81	±0.33 d–h	0.38	±0.20 a,b	8.48	±0.64 g	0.52	±0.01 i	0.14	±0.02 m	11.34	±0.90 j,k
		6	12.47	±0.36 b	2.44	±0.18 i,j	3.09	±0.15 j	2.9	±0.17 b–d	0.34	±0.04 g,h	0.11	±0.01 g–k	8.87	±0.46 f–i
	field	0	12.47	±0.36 b	4.13	±0.16 l	2.34	±0.2 i	0.21	±0.02 a	0.26	±0.01 e–g	0.09	±0.01 a–h	7.04	±0.28 c–f
		3	11.95	±0.66 b	1.82	±0.58 d–h	3.94	±0.53 k	6.06	±0.22 f	0.54	±0.05 i	0.08	±0.01 a–g	12.44	±0.93 k
		6	12.25	±0.52 b	3.77	±0.19 k,l	1.42	±0.11 e–h	2.47	±0.33 b–d	0.14	±0.01 a–e	0.08	±0.01 a–d	7.87	±0.15 d–h
Mean for A	Barcelona		11.93	±0.55 a	3.57	±2.53 c	2.73	±1.52 c	5.92	±5.81 c	0.15	±0.13 a	0.09	±0.02 a	12.45	±5.67 d
	Columbus		12.02	±0.29 a,b	2.65	±1.21 b	1.01	±0.60 a	4.64	±2.96 b	0.2	±0.14 b	0.09	±0.03 a,b	8.59	±3.53 b
	Strong Gold		12.12	±0.43 a,b	2.62	±1.76 b	1.12	±0.27 a	19.99	±19.56 d	0.13	±0.09 a	0.09	±0.16 a	23.94	±18.67 e
	Super Parrot		12.03	±0.30 a,b	1.52	±1.13 a	1.16	±0.79 a	2.17	±0.93 a	0.16	±0.11 a,b	0.1	±0.02 b	5.11	±2.45 a
	Tropicana		12.28	±0.40 b	2.61	±1.05 b	1.91	±1.41 b	4.8	±3.32 b	0.35	±0.17 c	0.1	±0.03 b	9.76	±2.16 c
Mean for B	greenhouse		12.03	±0.39 a	1.78	±0.82 a	1.12	±0.82 a	12.64	±12.22 b	0.22	±0.17 b	0.11	±0.02 b	15.86	±14.28 b
	field		12.12	±0.43 a	3.4	±2.01 b	2.05	±1.35 b	2.37	±1.25 a	0.17	±0.13 a	0.08	±0.01 a	8.08	±2.39 a
Mean for C	0		12.02	±0.46 a	2.78	±1.46 b	1.69	±1.34 b	6.35	±6.20 a	0.25	±0.14 c	0.1	±0.02 b	11.16	±5.93 a
	3		12.11	±0.38 a	2	±1.05 a	1.76	±1.40 b	9.29	±9.10 b	0.2	±0.19 b	0.1	±0.03 b	13.35	±11.87 b
	6		12.12	±0.39 a	2.99	±2.32 c	1.31	±0.76 a	6.87	±9.00 a	0.14	±0.09 a	0.09	±0.02 a	11.4	±11.30 a

* Means followed by the same letter are not significantly different at α = 0.05, using Duncan.

**Table 2 molecules-25-05627-t002:** Profiling of phenolic acids of tulip flowers [µg g^−1^ FW].

Cultivar (A)	Type of Cultivation (B)	Storage Length (Days) (C)	Gallic		Protocatechiuc		4-HBA		Vanilic		Syringic		Transcinnamin		2.5-DHBA	
Barcelona	greenhouse	0	0.49	±0.04 b–e *	2.99	±0.63 f	2.74	±0.41 i–l	2.15	±0.07 g–i	2.16	±0.33 b–e	22.71	±1.25 l	5.01	±0.36 q
		3	0.7	±0.09 d–f	0.71	±0.03 a,b	3.83	±0.40 no	1.5	±0.35 d–f	3.46	±0.74 f,g	29.17	±1.84 n	2.68	±0.44 l–n
		6	1.38	±0.18 j,k	2.07	±0.12 c–f	2.11	±0.18 e–h	1.06	±0.09 b–e	4.16	±0.36 g–i	4.56	±0.56 d–g	1.88	±0.38 j,k
	field	0	1.12	±0.20 i,j	1.6	±0.18 b–d	4.28	±0.49 o	2.64	±0.38 i–k	1.06	±0.19 a	6.36	±0.35 f–i	2.44	±0.25 l,m
		3	1.12	±0.14 i,j	7.38	±0.45 i,j	6.69	±0.47 q	1.53	±0.40 d–f	5.82	±0.91 k	3.02	±0.27 b–d	1.51	±0.45 e–j
		6	0.82	±0.03 f–h	8.51	±0.36 j	1.3	±0.27 b–d	1.67	±0.11 f,g	3.96	±0.09 g–i	3.7	±0.33 c,d	3.35	±0.22 o
Columbus	greenhouse	0	2.55	±0.30 m	7	±1.55 i	2.35	±0.39 f–j	2.09	±0.50 g,h	2.24	±0.62 c–e	26.9	±3.54 m	1.33	±0.15 d–i
		3	1.64	±0.06 k	5.01	±0.45 h	2.95	±0.29 j–m	3.72	±0.40 l	7.35	±0.96 l	22.05	±1.75 l	2.33	±0.45 k,l
		6	0.49	±0.08 b–e	7.03	±0.19 i	2.92	±0.19 j–l	3.1	±0.28 k	7.47	±0.45 l	22.34	±2.11 l	0.61	±0.01 a,b
	field	0	0.78	±0.06 e–g	7.5	±1.48 i,j	1.81	±0.15 d–f	5.04	±0.50 m	4.07	±0.11 g–i	1.13	±0.19 a,b	1.38	±0.14 d–j
		3	0.74	±0.04 e–g	0.51	±0.15 a	1.34	±0.18 b–d	1.48	±0.33 d–f	1.28	±0.39 a,b	1.31	±0.22 a,b	1.86	±0.60 i,j
		6	0.32	±0.02 a–c	1.29	±0.28 a–c	2.2	±0.40 e–i	3.85	±0.32 l	3.69	±0.59 g–i	1.5	±0.22 a–c	0.76	±0.12 a–c
Strong Gold	greenhouse	0	2.14	±0.10 l	4.95	±0.52 h	1.91	±0.24 d–g	4.07	±0.25 l	2.56	±0.29 d,e	12.38	±1.33 k	2.83	±0.33 m,n
		3	0.42	±0.08 a–d	1.42	±0.24 a–c	0.88	±0.15 a,b	1.65	±0.10 f,g	5.66	±0.40 k	7.14	±0.59 h,i	0.91	±0.28 b–d
		6	3.06	±0.27 n	2.04	±0.37 c–e	1.47	±0.34 b–d	0.32	±0.03 a	5.29	±0.66 j,k	3.02	±0.16 b–d	1.41	±0.29 d–j
	field	0	2.84	±0.20 n	5.03	±0.32 h	3.13	±0.78 k–m	2.71	±0.51 j,k	3.6	±0.23 g,h	4.34	±0.16 d–g	4.1	±0.18 p
		3	2.94	±0.06 n	1.33	±0.16 a–c	1.12	±0.18 a–c	2.49	±0.35 h–j	1.64	±0.12 a–d	1.58	±0.33 a–c	1.68	±0.08 g–j
		6	0.31	±0.04 a–c	1.68	±0.24 b–d	1.71	±0.08 c–e	1.49	±0.18 d–f	4.47	±0.41 h–j	2.62	±0.25 a–d	1.63	±0.19 f–j
Super Parrot	greenhouse	0	0.37	±0.05 a–c	0.92	±0.25 a,b	0.6	±0.07 a	1.01	±0.30 b–d	1.41	±0.08 a–c	3.14	±0.52 b–d	0.41	±0.04 a
		3	5.07	±0.21 o	2.9	±0.34 e,f	0.54	±0.05 a	0.54	±0.12 a,b	4.02	±0.66 g–i	1.66	±0.25 a–c	1.69	±0.22 g–j
		6	2.19	±0.41 l	0.86	±0.36 a,b	1.14	±0.17 a–c	2.09	±0.14 g,h	1.83	±0.08 a–e	0.67	±0.08 a	1.11	±0.28 c–f
	field	0	0.21	±0.01 a,b	2.4	±0.32 d–f	1.6	±0.26 c–e	0.79	±0.25 a–c	2.65	±0.55 e,f	10.02	±2.52 j	2.37	±0.09 l,m
		3	0.53	±0.02 c–f	1.19	±0.17 a–c	3.33	±0.31 l–n	1.13	±0.11 c–f	2.39	±0.38 d,e	4.57	±0.23 d–g	1.32	±0.23 d–h
		6	0.14	±0.01 a	1.24	±0.16 a–c	0.96	±0.11 a,b	0.92	±0.07 b,c	4.35	±0.17 g–i	8.36	±1.04 i,j	1.73	±0.09 h–j
Tropicana	greenhouse	0	0.34	±0.06 a–c	9.15	±0.70 k	3.53	±0.39 m,n	1.28	±0.13 c–f	4.29	±0.72 g–i	35.75	±2.94 o	0.35	±0.12 a
		3	0.55	±0.02 c–f	4.97	±0.17 h	0.98	±0.22 a,b	1.54	±0.35 d–f	4.58	±0.27 i,j	4.1	±0.38 d–f	1.18	±0.24 c–g
		6	1	±0.08 g–i	7.13	±0.29 i	2.53	±0.35 h–k	3.11	±0.41 k	6.9	±0.32 l	6.01	±0.35 e–h	0.54	±0.02 a,b
	field	0	1.1	±0.18 h–j	3.97	±0.29 g	6.75	±0.36 q	8.38	±0.51 n	7.36	±0.33 l	6.44	±0.36 g–i	9.35	±0.38 r
		3	2.98	±0.47 n	2.44	±0.40 d–f	5.06	±0.51 p	4.05	±0.14 l	9.24	±0.63 m	2.57	±0.20 a–d	2.93	±0.11 n,o
		6	0.47	±0.04 b–e	1.63	±0.46 b–d	2.49	±0.44 g–j	1.6	±0.16 e–g	11.61	±0.89 n	3.77	±0.57 c–e	1	±0.28 b–e
Mean for A	Barcelona		0.94	±0.33 a	3.83	±2.99 c	3.49	±1.82 d	1.76	±0.57 b	3.44	±1.62 b	11.59	±10.70 c	2.81	±1.22 d
	Columbus		1.09	±0.81 b	4.72	±3.00 d	2.26	±0.64 c	3.21	±1.25 d	4.35	±2.47 d	12.54	±11.80 d	1.38	±0.67 a
	Strong Gold		1.95	±1.20 d	2.74	±1.68 b	1.7	±0.81 b	2.12	±1.22 c	3.87	±1.51 c	5.18	±3.81 a	2.09	±1.12 b
	Super Parrot		1.42	±1.34 c	1.59	±2.70 d	1.36	±0.99 a	1.08	±0.53 a	2.77	±1.16 a	4.74	±3.64 a	1.44	±0.64 a
	Tropicana		1.07	±0.94 b	4.88	±0.83 a	3.56	±1.97 d	3.33	±2.55 d	7.32	±2.66 e	9.78	±9.10 b	2.56	±2.25 c
Mean for B	greenhouse		1.49	±1.31 b	3.94	±2.70 b	2.03	±1.06 a	1.95	±1.12 a	4.23	±2.00 a	13.44	±11.59 b	1.62	±1.22 a
	field		1.09	±0.98 a	3.16	±2.59 a	2.92	±1.93 b	2.65	±1.99 b	4.48	±2.92 b	4.09	±2.68 a	2.49	±2.06 b
Mean for C	0		1.19	±0.95 b	4.55	±2.68 c	2.87	±1.69 c	3.01	±2.25 b	3.14	±1.79 a	12.92	±11.31 c	2.96	±2.61 c
	3		1.67	±1.49 c	2.79	±2.22 a	2.67	±2.01 b	1.96	±1.11 a	4.54	±2.50 b	7.72	±7.42 b	1.81	±0.70 b
	6		1.02	±0.92 a	3.32	±2.80 b	1.88	±0.68 a	1.92	±1.09 a	5.37	±2.65 c	5.66	±5.08 a	1.4	±0.82 a
**Cultivar** **(A)**	**Type of Cultivation** **(B)**	**Storage Length (Days)** **(C)**	**Caffeic**		**p–Cumaric**		**Chlorogenic**		**Ferulic**		**Sinapic**		**Salicylic**		**Sum**	
Barcelona	greenhouse	0	4.24	±0.20 j	0.62	±0.07 d–h	1.77	±0.25 a–c	2.24	±0.35 h	13.26	±1.88 k	0.14	±0.15 a	60.53	±0.50 k
		3	1.42	±0.19 d–f	2.06	±0.14 n	0.17	±0.01 a	1.68	±0.33 f,g	7.02	±0.28 j	0.15	±0.04 a	54.55	±3.69 j
		6	0.64	±0.02 a,b	0.67	±0.08 e–h	1.35	±0.24 a,b	1.49	±0.06 d–f	13.74	±1.44 k	0.73	±0.19 a–c	35.84	±1.49 g
	field	0	3.47	±0.47 i	0.4	±0.06 b–e	0.97	±0.13 a,b	1.15	±0.20 c–f	4.79	±0.23 g,h	0.62	±0.12 a–c	30.92	±1.41 e,f
		3	0.55	±0.12 a	1.55	±0.47 m	1.44	±0.18 a,b	0.34	±0.04 a,b	16.92	±1.50 l	0.1	±0.01 a	47.97	±2.18 i
		6	2.35	±0.33 g	1.28	±0.27 k–m	0.17	±0.02 a	2	±0.37 g,h	8.16	±0.54 j	0.63	±0.04 a–c	37.61	±0.78 g
Columbus	greenhouse	0	0.81	±0.14 a–c	0.37	±0.06 a–e	32.98	±5.25 e	0.74	±0.23 b,c	3.23	±0.60 c–g	0.29	±0.02 a,b	82.89	±5.48 m
		3	1.21	±0.38 c–f	1.24	±0.13 j–k	1.56	±0.48 a,b	1.42	±0.34 d–f	4.64	±0.27 g,h	0.45	±0.03 a,b	55.58	±2.56 j
		6	0.62	±0.03 a,b	0.34	±0.06 a–d	1.62	±0.21 a,b	1.25	±0.15 c–f	3.98	±0.28 e–h	8.78	±0.78 i	60.55	±3.08 k
	field	0	0.62	±0.06 a,b	0.91	±0.03 h,i	3.51	±0.44 c	0.76	±0.10 b,c	13.39	±2.12 k	2.18	±0.15 e	43.08	±4.25 h
		3	0.55	±0.13 a	0.16	±0.02 a,b	0.38	±0.13 a,b	1.03	±0.22 c–e	15.42	±1.65 l	0.67	±0.12 a–c	26.73	±1.92 c–e
		6	1.62	±0.23 f	1.45	±0.33 l,m	0.46	±0.03 a,b	3.32	±0.10 i	4.8	±0.78 g,h	0.79	±0.03 a–c	26.05	±1.21 b–d
Strong Gold	greenhouse	0	5.17	±0.36 k	0.57	±0.05 c–f	0.92	±0.06 a,b	0.94	±0.03 c,d	4.96	±0.06 g–i	0.2	±0.03 a	43.61	±2.83 h
		3	1.56	±0.38 e,f	0.41	±0.08 b–e	0.17	±0.00 a	0.97	±0.10 c,d	4.08	±0.43 e–h	0.23	±0.06 a	25.49	±0.22 b–d
		6	0.39	±0.04 a	3.56	±0.30 p	0.15	±0.11 a	1.12	±0.08 c–e	3.86	±0.43 e–h	1.17	±0.14 c,d	26.85	±0.71 c–e
	field	0	1.03	±0.22 b–d	0.99	±0.21 i,j	0.92	±0.15 a,b	1.21	±0.13 c–f	4.61	±0.3 g,h	0.61	±0.04 a–c	35.11	±2.28 f,g
		3	2.6	±0.43 g,h	0.8	±0.07 f–i	3.38	±0.35 c	1.17	±0.10 c–f	2.42	±0.45 b–e	0.96	±0.06 b–d	24.12	±1.63 b,c
		6	2.85	±0.10 h	0.18	±0.02 a,b	1.56	±0.24 ab	1.56	±0.40 e–g	3.39	±0.37 c–h	0.39	±0.04 a,b	23.83	±0.94 b,c
Super Parrot	greenhouse	0	0.51	±0.09 a	3.05	±0.18 o	0.14	±0.03 a	1.46	±0.37 d–f	0.97	±0.2 a,b	3.56	±0.31 g	17.54	±1.37 a
		3	2.26	±0.55 g	0.09	±0.82 a	1.08	±0.62 a,b	1.19	±2.68 c–f	51	±3.63 a	0.26	±1.38 a	21.81	±1.52 b
		6	2.5	±0.30 g,h	0.53	±0.07 c–f	0.82	±0.24 a,b	0.15	±0.05 a	0.29	±0.01 a	1.31	±0.15 c,d	15.5	±0.60 a
	field	0	0.52	±0.06 a	0.16	±0.03 a,b	0.54	±0.05 a,b	0.41	±0.07 a,b	5.31	±0.14 h,i	1.3	±0.31 c,d	28.28	±2.42 c–e
		3	1.35	±0.10 d–f	1.36	±0.11 l,m	2.29	±0.28 b,c	1.21	±0.15 c–f	1.67	±0.59 a–c	2.23	±0.39 e	24.55	±2.00 b–d
		6	0.85	±0.10 a–c	0.88	±0.07 g–i	0.91	±0.07 a,b	0.72	±0.03 b,c	4.38	±0.26 f–h	1.52	±0.13 d	26.95	±1.15 c–e
Tropicana	greenhouse	0	1.13	±0.08 c–e	0.53	±0.04 c–f	0.14	±0.01 a	7.35	±0.81 j	2.67	±0.28 b–f	0.19	±0.00 a	66.69	±5.17 l
		3	0.55	±0.12 a	0.3	±0.02 a–c	0.13	±0.01 a	8.13	±0.61 k	1.9	±0.53 a–d	5.81	±0.25 h	34.72	±1.45 f,g
		6	1.39	±0.18 d–f	0.55	±0.10 c–f	6.81	±0.48 d	1.15	±0.15 c–f	3.7	±0.58 d–h	10.8	±1.65 j	51.64	±3.74 i,j
	field	0	3.5	±0.25 i	0.61	±0.03 d–g	1.35	±0.07 a,b	1.09	±0.17 c–e	16.48	±2.91 l	2.83	±0.17 f	69.2	±4.05 l
		3	3.86	±0.32 i,j	1.04	±0.27 i–k	1.4	±0.20 a,b	1.18	±0.18 c–f	6.64	±1.63 i,j	0.25	±0.02 a	43.65	±1.04 h
		6	1.51	±0.24 e,f	1.01	±0.09 i–j	1.49	±0.40 a,b	1.22	±0.14 c–f	1.02	±0.18 a,b	0.33	±0.02 a,b	29.16	±1.41 d,e
Mean for A	Barcelona		2.11	±1.44 c,d	1.1	±0.63 b	0.98	±0.65 a	1.48	±0.68 c	10.65	±4.50 e	0.39	±0.29 a	44.56	±11.10 c
	Columbus		0.9	±0.42 a	0.75	±0.52 a	6.75	±12.3 c	1.42	±0.93 c	7.58	±5.13 d	2.19	±2.11 c	49.15	±20.70 d
	Strong Gold		2.27	±1.67 d	1.09	±1.10 b	1.18	±1.14 a	1.16	±0.26 b	3.89	±0.91 b	0.59	±0.38 a	29.84	±7.58 b
	Super Parrot		1.33	±0.85 b	1.01	±1.00 b	0.96	±0.70 a	0.86	±0.52 a	2.19	±2.02 a	1.7	±1.07 b	22.44	±5.01 a
	Tropicana		1.99	±1.29 c	0.68	±0.29 b	1.89	±1.35 b	3.35	±3.22 d	5.4	±5.30 c	3.37	±3.04 d	49.18	±15.70 d
Mean for B	greenhouse		1.63	±1.39 a	0.99	±1.03 b	3.32	±3.26 b	2.09	±2.01 b	4.59	±3.98 a	2.27	±2.20 b	43.59	±19.80 b
	field		1.81	±0.74 b	0.53	±0.48 a	1.38	±0.99 a	1.23	±0.71 a	7.29	±5.47 b	1.03	±0.81 a	34.48	±32.22 a
Mean for C	0		2.1	±1.74 b	0.82	±0.80 a	4.33	±4.26 b	1.74	±1.70 b	6.97	±5.27 c	1.19	±1.15 a	47.79	±20.51 c
	3		1.59	±1.05 a	0.9	±0.66 a	1.2	±1.04 a	1.83	±1.73 b	6.12	±5.56 b	1.11	±1.01 a	35.92	±12.97 b
	6		1.47	±0.85 a	1.05	±0.95 b	1.53	±1.44 a	1.4	±0.83 a	4.73	±3.71 a	2.64	±2.51 b	33.4	±13.20 a

* Means followed by the same letter are not significantly different at α = 0.05, using Duncan.

**Table 3 molecules-25-05627-t003:** Profiling of organic acids of tulip flowers [µg g^−1^ FW].

Cultivar (A)	Type of Cultivation (B)	Storage Length (Days) (C)	Oxalic		Quinic		Malic		Malonic		Citric	
Barcelona	greenhouse	0	0.02	±0.05 a *	0.19	±0.04 a,b	0	±0.00 a	7.53	±1.48 b–e	0	±0.00 a
		3	0.01	±0.00 a	0.28	±0.10 a–d	0	±0.00 a	3.92	±0.61 a–d	0	±0.00 a
		6	0.12	±0.16 a–c	0.34	±0.15 b–e	0	±0.00 a	6.01	±0.30 a–e	0.2	±0.04 a
	field	0	0.04	±0.00 a,b	0.3	±0.09 a–d	0	±0.00 a	11.61	±2.40 e–g	19.99	±4.38 e
		3	0.08	±0.01 a,b	2.8	±0.41 i	0	±0.00 a	73.96	±8.89 i	0.27	±0.01 a
		6	0.04	±0.01 a,b	0.52	±0.02 c–f	0.42	±0.02 c	8.13	±0.70 c–e	0.06	±0.00 a
Columbus	greenhouse	0	0.01	±0.00 a	0.32	±0.15 b–e	0	±0.00 a	7.38	±2.13 b–e	0	±0.00 a
		3	0.02	±0.01 a	0.37	±0.06 b–e	0	±0.00 a	8.91	±2.77 d–f	7.76	±0.82 c
		6	0.02	±0.00 a	0.29	±0.09 a–d	0	±0.00 a	7.46	±3.05 b–e	0.15	±0.04 a
	field	0	0.05	±0.02 a,b	0.67	±0.36 f	0	±0.00 a	10.14	±3.33 d–f	0	±0.00 a
		3	0.05	±0.01 a,b	1.73	±0.25 h	0	±0.00 a	1.86	±0.22 a–c	12.55	±1.62 d
		6	0.24	±0.30 c	0.3	±0.03 a–d	0	±0.00 a	1.5	±0.26 a,b	0.19	±0.14 a
Strong Gold	greenhouse	0	0.02	±0.01 a	0.22	±0.04 a–c	0	±0.00 a	15.26	±3.33 f–h	0.13	±0.04 a
		3	0.02	±0.01 a	0.18	±0.07 a,b	0	±0.00 a	8.92	±0.52 d–f	0.11	±0.02 a
		6	0.01	±0.00 a	0.47	±0.25 b–f	0	±0.00 a	19.52	±8.21 h	1.06	±0.21 a,b
	field	0	0.12	±0.15 a–c	0.35	±0.08 b–e	0	±0.00 a	18.95	±2.54 h	13.92	±3.22 d
		3	0.03	±0.00 a	0.28	±0.08 a–d	0.2	±0.01 b	1.73	±0.18 a–c	24.31	±2.31 f
		6	0.01	±0.00 a	0.26	±0.01 a–d	0	±0.00 a	0.27	±0.04 a	0	±0.00 a
Super Parrot	greenhouse	0	0.05	±0.01 a,b	0.22	±0.03 a–c	0	±0.00 a	9.09	±4. d–f 07	1.46	±0.67 a,b
		3	0.04	±0.03 a,b	0.28	±0.11 a–d	0	±0.00 a	9.44	±4.43 d–f	1.17	±1.11 a,b
		6	0.04	±0.14 a,b	0.24	±0.08 a–c	0	±0.00 a	7.44	±4.50 b–e	3.67	±4.24 b
	field	0	0.08	±0.01 a,b	1.3	±0.14 g	1.07	±0.05 d	8.27	±2.79 c–e	14.44	±1.85 d
		3	0.03	±0.01 a,b	0.56	±0.04 d–f	0	±0.00 a	14.96	±1.65 f–h	0.1	±0.08 a
		6	0.03	±0.01 a,b	1.32	±0.34 g	0	±0.00 a	2.19	±0.10 a–c	21.99	±1.06 e,f
Tropicana	greenhouse	0	0.02	±0.01 a	0	±0.00 a	0	±0.00 a	5.16	±4.71 a–e	2.48	±0.73 a,b
		3	0.02	±0.01 a	0.24	±0.06 a–c	0	±0.00 a	4.85	±2.68 a–d	0.19	±0.06 a
		6	0.02	±0.00 a	0.32	±0.02 b–e	0	±0.00 a	100.03	±6.24 j	0	±0.00 a
	field	0	0.17	±0.02 b,c	1.57	±0.29 g,h	0	±0.00 a	17.13	±3.50 g,h	0	±0.00 a
		3	0.06	±0.01 a,b	0.31	±0.04 a–e	0	±0.00 a	0.15	±0.01 a	0	±0.00 a
		6	0.08	±0.01 a,b	0.62	±0.02 e,f	0	±0.00 a	1.67	±0.32 a–c	8.6	±0.52 c
Mean for A	Barcelona		0.05	±0.04 a	0.74	±0.64 d	0.07	±0.01 c	18.53	±12.58 c	3.42	±2.77 b
	Columbus		0.06	±0.03 a	0.61	±0.56 b,c	0	±0.00 a	6.21	±3.96 a	3.44	±3.14 b
	Strong Gold		0.04	±0.02 a	0.29	±0.14 a	0.03	±0.00 b	10.77	±8.57 b	6.59	±5.73 c
	Super Parrot		0.05	±0.02 a	0.65	±0.51 c,d	0.18	±0.02 d	8.57	±4.77 b	7.14	±6.16 c
	Tropicana		0.06	±0.05 a	0.51	±0.48 b	0	±0.00 a	21.5	±20.10 d	1.88	±3.24 a
Mean for B	greenhouse		0.03	±0.05 a	0.26	±0.13 a	0	±0.00 a	14.73	±13.59 b	1.23	±1.28 a
	field		0.07	±0.09 b	0.86	±0.74 b	0.11	±0.00 b	11.5	±11.16 a	7.76	±7.17 b
Mean for C	0		0.06	±0.06 a	0.51	±0.52 a	0.11	±0.02 c	11.05	±5.06 a	5.24	±5.20 b
	3		0.04	±0.02 a	0.7	±0.85 b	0.02	±0.00 a	12.87	±11.36 b	4.65	±4.01 b
	6		0.06	±0.11 a	0.47	±0.34 a	0.04	±0.01 b	15.42	±19.34 c	3.59	±3.30 a
**Cultivar** **(A)**	**Type of Cultivation** **(B)**	**Storage Length (Days) (C)**		**Acetic**	**Fumaric**			**Succinic**		**Sum**		
Barcelona	greenhouse	0	19.44	±7.06 c–e	0.09	±0.03 a–d		23.46	±2.91 e–i	50.73	±11.43 e–j	
		3	23.03	±11.94 d–f	0.18	±0.13 a–d		11.34	±1.90 a–f	38.77	±9.65 b–f	
		6	13.12	±1.52 b–d	0.16	±0.05 a–d		14.98	±0.83 a–h	34.94	±2.61 a–f	
	field	0	1.01	±0.14 a	0.22	±0.07 a–e		30.97	±6.55 g–i	64.14	±4.84 h–j	
		3	136.41	±24.6 j	0.06	±0.04 a–d		65.7	±3.29 j	279.29	±19.80 l	
		6	52.86	±2.70 h	0.26	±0.05 d,e		0.26	±0.05 a,b	62.55	±1.97 g–j	
Columbus	greenhouse	0	8.19	±5.58 a,b	0	±0.00 a		18.67	±10.05 c–h	34.56	±6.39 a–f	
		3	0	±0.00 a	0	±0.00 a		18.77	±14.72 c–h	35.83	±16.62 a–f	
		6	5.12	±0.89 a,b	0.01	±0.00 a–c		7.85	±2.81 a–e	20.9	±1.62 a,b	
	field	0	33.86	±4.81 g	0.14	±0.04 a–d		23.75	±10.10 e–i	68.6	±12.20 j	
		3	0	±0.00 a	0.24	±0.04 c–e		26.53	±2.31 f–i	42.96	±1.14 b–h	
		6	28.17	±2.21 e–g	0.1	±0.00 a–d		0	±0.00 a	30.5	±2.59 a–f	
Strong Gold	greenhouse	0	30.25	±4.49 f,g	0.2	±0.03 a–e		20.78	±4.08 d–h	66.87	±11.66 j	
		3	10.72	±2.86 a–c	0.23	±0.05 b–e		12.53	±3.12 a–f	32.71	±5.41 a–f	
		6	9.62	±2.02 a–c	0.17	±0.15 a–d		13.63	±1.23 a–g	44.48	±9.41 c–i	
	field	0	4.4	±0.61 a,b	0.22	±0.03 a–e		27.7	±4.18 f–i	65.65	±4.56 i,j	
		3	0	±0.00 a	0.14	±0.01 a–d		22.33	±2.14 d–h	49.03	±4.22 e–j	
		6	21.84	±0.98 d–f	0.4	±0.51 e		0.3	±0.04 a,b	23.09	±1.44 a–c	
Super Parrot	greenhouse	0	3.9	±3.37 a,b	0	±0.00 a		1.68	±0.18 a–c	16.4	±7.35 a	
		3	2.2	±1.55 a,b	0.02	±0.01 a–c		23.1	±21.20 d–h	36.26	±32.76 a–f	
		6	1.2	±0.66 a	0.02	±0.01 a–c		39.77	±25.20 i	52.38	±29.70 f–j	
	field	0	1.28	±0.32 a	0.06	±0.04 a–d		13.65	±2.05 a–g	40.16	±2.88 b–f	
		3	0	±0.00 a	0.01	±0.00 a,b		32.18	±5.65 h,i	47.84	±7.12 d–j	
		6	0	±0.00 a	0	±0.00 a		90.47	±5.92 k	116	±5.65 k	
Tropicana	greenhouse	0	6.62	±2.53 a,b	0.08	±0.05 a–d		14.93	±9.00 a–h	29.29	±10.81 a–e	
		3	3.31	±1.66 a,b	0.07	±0.02 a–d		6.66	±0.62 a–e	15.35	±4.74 a	
		6	3.09	±0.18 a,b	0.03	±0.00 a–c		5.96	±0.30 a–d	109.44	±6.69 k	
	field	0	77.06	±7.26 i	0.15	±0.06 a–d		0.88	±0.06 a,b	96.95	±11.1 k	
		3	23.46	±1.02 d–f	0.12	±0.03 a–d		17.57	±1.37 b–h	41.67	±1.50 b–g	
		6	0	±0.00 a	0.13	±0.02 a–d		15.11	±0.57 a–h	26.2	±0.84 a–d	
Mean for A	Barcelona		40.98	±30.8 d	0.16	±0.09 c,d		24.45	±21.6 b	88.4	±80.00 c	
	Columbus		12.56	±12.10 b	0.08	±0.05 a,b		15.93	±11.9 a	38.89	±17.00 a	
	Strong Gold		12.81	±0.21 b	0.23	±0.21 d		16.21	±9.36 a	46.97	±17.50 b	
	Super Parrot		1.43	±10.80 a	0.02	±0.01 a		33.48	±32.3 c	51.51	±35.60 b	
	Tropicana		18.92	±16.00 c	0.1	±0.07 b,c		10.18	±6.95 a	53.15	±37.90 b	
Mean for B	greenhouse		9.32	±8.30 a	0.09	±0.10 a		15.61	±13.26 a	41.26	±19.80 a	
	field		25.36	±12.22 b	0.15	±0.15 b		24.49	±24.22 b	70.31	±12.22 b	
Mean for C	0		18.61	±15.19 b	0.12	±0.09 a		17.65	±11.00 a	53.34	±20.51 a	
	3		19.91	±20.01 b	0.11	±0.10 a		23.67	±18.52 b	61.97	±12.97 b	
	6		13.5	±12.19 a	0.13	±0.11 a		18.83	±17.69 a	52.05	±13.20 a	

* Means followed by the same letter are not significantly different at α = 0.05, using Duncan.

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
