# Peer review of "Content of Phenolic Compounds and Organic Acids in the Flowers of Selected Tulipa gesneriana Cultivars"

_molecules, 2020, doi:10.3390/molecules25235627_

Round 1

Reviewer 1 Report

This study looks at the phenolic, flavonoids and organic acids from 5 tulip cultivators-Barcelona, Columbus, Strong gold, Super parrot and Tropicana.

Abstract: It does not address the significance of this study. The introduction states that the data from this study would help to determine which cultivar is rich in flavonoids and organic acids which are beneficial to human health and could be included in food.

Introduction:

  1. Line 33-35 “Phenolic compounds are secondary metabolites recognised in edible flowers as bioactive compounds and the most important antioxidants responsible for anti-inflammatory, anticarcinogenic and antibacterial properties and determining the colour of flower”

Too long sentence and needs to be broken down.

  1. Line 49-50 Thanks to their biochemical content, tulip flowers exhibit biological activities, such as antipyretic, anticancer, laxative, expectorant and depurative.

Specify which compounds have these medicinal value and the name of the cultivars. If they are same cultivars then what is unique about this study?

Results:

  1. HPLC standards were used to know the “area under the curve” that determined the amounts of phenols or organic acids. These results were not affirmed by thin layer chromatography or any biochemical assay to see if the compounds that matched the standard was indeed the compound. The quantification seems very crude and needs more confirmation tests to make substantial conclusions.
  2. Line 159 This study is a novel study on the chemical characteristics of this plant. Compare and contrast with the tulip cultivars that have been studied before that makes the organic acids found in the five cultivars unique.
  3. The phenolic and flavonoid content can be verified for its activity by doing a radical scavenging assay.
  4. Previous studies on tulips have shown other phenolic such as populentin. This was not analyzed here. What was the basis to select and analyze only couple of phenolics and organic acids?

Methods

3.3 Write the names of the chemicals phenols and organic acids standards used. What kind of HPLC was used?

Author Response

Thank You for all remarks. We consulted they. Our responses are in attached file.

Reviewer 2 Report

It was interesting to read the manuscript by Krzymińska and colleagues "Content of phenolic compounds and organic acids in 3 the flowers of selected Tulipa gesneriana cultivars", as the concept of using tulip petals in human nutrition was novel to me.

Nevertheless, the significance of the results is rather low, because the contribution of this plant material in dietary intake of health-promoting compounds is very low, and will most likely never be inculded in everyday diet.

The experimental setting was planned well, and the methods are appropriate. I would like to see a visual presentation of the results rather than the tables; it was difficult to follow the results in table format. As there were three replicates for each experimental combination, the authors should also show the standard errors of the results.

In Discussion and Conclusions, I would like to see more discussion on the potential of tulip petals in nutrition compared to other edible plants.

Author Response

(The authors gave the same response as above.)
